# Super-diffusion of excited carriers in semiconductors

Ebrahim Najafi[1], Vsevolod Ivanov[2,3], Ahmed Zewail[1,‡] & Marco Bernardi[2]

The ultrafast spatial and temporal dynamics of excited carriers are important to understanding the response of materials to laser pulses. Here we use scanning ultrafast electron microscopy to image the dynamics of electrons and holes in silicon after excitation with a short laser pulse. We find that the carriers exhibit a diffusive dynamics at times shorter than 200 ps, with a transient diffusivity up to 1,000 times higher than the room temperature value, $D_0 \approx 30 \, \mathrm{cm^2 s^{-1}}$. The diffusivity then decreases rapidly, reaching a value of $D_0$ roughly 500 ps after the excitation pulse. We attribute the transient super-diffusive behaviour to the rapid expansion of the excited carrier gas, which equilibrates with the environment in $100 - 150 \, \mathrm{ps}$. Numerical solution of the diffusion equation, as well as *ab initio* calculations, support our interpretation. Our findings provide new insight into the ultrafast spatial dynamics of excited carriers in materials.

[1] Physical Biology Center for Ultrafast Science and Technology, Arthur Noyes Laboratory of Chemical Physics, California Institute of Technology, Pasadena, California 91125, USA. [2] Department of Applied Physics and Materials Science, Steele Laboratory, California Institute of Technology, Pasadena, California 91125, USA. [3] Department of Physics, California Institute of Technology, Pasadena, California 91125, USA. Correspondence and requests for materials should be addressed to M.B. (email: bmarco@caltech.edu).
‡Deceased

Ultrafast lasers have enabled detailed studies of the dynamics of charge carriers in materials in the fs − ns time domain[1,2]. Yet, the ultrafast spatial dynamics of excited carriers is still challenging to characterize. For example, pump-probe microscopy with visible light is inherently limited by diffraction, and it remains a non-routine approach in spite of recent progress[3]. Studies of the spatiotemporal evolution of excited carriers have recently become possible using scanning ultrafast electron microscopy (SUEM)[4–7], a technique that combines the spatial resolution of electron microscopy and the time resolution of ultrafast lasers.

Knowledge of the spatial distribution of excited carriers provides a more complete understanding of ultrafast carrier dynamics in materials. For example, exciting a silicon p-n junction with a laser[5] induces a so-called super-diffusive transport regime in which the distance covered by excited carriers over a time $\tau$ is significantly longer than the diffusion length $L_D = \sqrt{D_0 \tau}$., where $D_0$ is the room temperature carrier diffusivity. Similarly, ultrafast laser-induced demagnetization of ferromagnetic films has been explained using super-diffusion of charge carriers and spin, but this process is still the subject of considerable research to unravel its mechanism[8,9].

Theoretical modelling of excited carrier dynamics is still in its infancy. First-principles calculations have recently been employed to study carrier dynamics in the fs − ps time domain for spatially uniform carrier distributions in semiconductors and metals[10–14]. Yet, modelling from first principles the spatial dynamics of carriers on the μm length scale still constitutes a formidable challenge. On the μm length scale, the dynamics of excited carriers typically occurs in the hydrodynamic regime[15], and it can often be modelled satisfactorily through semiclassical transport theory[16,17], as is done in this work.

Here, we measure the spatiotemporal evolution of the electron and hole populations after laser excitation in silicon using the SUEM technique[4,5]. Analysis of the second moment $\langle R^2 \rangle$ of the carrier spatial distribution reveals two distinct regimes for the carrier dynamics. A transient super-diffusive dynamics is found for times shorter than ∼ 200 ps, where the carriers exhibit a transient diffusivity $D \propto \partial \langle R^2 \rangle / \partial t$ larger by a factor of ∼ 1,000 than the room temperature diffusivity[18] and monotonically increasing with the laser fluence. For times longer than ∼ 500 ps, the carriers exhibit a typical steady-state diffusive transport regime, with diffusivity $D_0 \approx 30 \, \text{cm}^2 \, \text{s}^{-1}$ (ref. 18). Treating the excited carriers as a hot non-degenerate electron gas, we attribute the rapid transient expansion to the large pressure gradient in the carrier population induced by the laser pulse. We derive and solve numerically a diffusion equation for an initial gaussian carrier concentration profile, employing a diffusivity that decays exponentially in time as a result of the cooling of the excited carriers. Excellent agreement with the experimental results is found. Taken together, we demonstrate the existence of a transient carrier super-diffusive regime in semiconductors, and explain its origin. Our first direct observation of super-diffusion advances understanding of the excited carrier spatial dynamics in materials.

## Results

**Imaging the carrier dynamics**. We image the carrier dynamics in samples of p-type and n-type silicon using SUEM. Briefly, the samples are excited with a 515 nm wavelength gaussian laser pulse with a fluence in the 0.16 − 1.28 mJ cm$^{-2}$ range. Electron pulses with a controlled time delay between − 760 ps and 3.32 ns raster-scan the surface, generating secondary electrons (SEs) which are then collected to form images. The time and space resolutions are, respectively, 2 ps and 200 nm. To enhance the

signal, the background is removed by subtracting a reference image recorded before optical excitation (at − 760 ps) from the images recorded at different time delays. This approach yields 'contrast images', in which the bright and dark contrasts are interpreted as increased electron and hole concentrations, respectively. Since each excited carrier has a given cross section for generating SEs during its lifetime, the total rate of SE emission is expected to increase linearly with carrier concentration. This result is consistent with the increased intensity of the SUEM images observed for increasing fluence values. For a given material, the number of emitted SEs also depends on the surface topography, chemical composition, and local fields[19]. Removal of the background ensures that the observed contrast reflects only the changes in local carrier density due optical excitation. A schematic of the experiment is shown in Fig. 1, and experimental details are given in the Methods section.

Figure 2 shows contrast images recorded for our n-type and p-type silicon samples. Because the samples are highly doped, a large excess of excited holes (electrons) are generated in n-type (p-type) silicon upon laser excitation. The contrast images thus represent the evolution of excess electrons and holes in p-type and n-type silicon, respectively. A bright contrast is seen for excess electrons (Fig. 2a), and a dark contrast for excess holes (Fig. 2b), both mapping the spatial distribution of excited carriers. These images, collected at several time delays, are used to construct movies of the carrier dynamics (see Supplementary Movies 1 and 2 for electron and hole dynamics, respectively).

Following the optical excitation, the contrast images for p-type silicon (Fig. 2a) show a bright spot, the intensity of which increases monotonically and peaks at a delay time of t = 50 ps. The bright contrast reflects the increased electron concentration near the surface, with a gaussian profile at t = 50 ps closely resembling that of the laser intensity on the surface. We attribute the brightness increase up to 50 ps after excitation to a redistribution of the excited electron energy due to ultrafast electron-electron and electron-phonon scattering processes[2,10]. After generating the initial population through a monochromatic excitation, the energy distribution of the excited electrons becomes broader due to scattering, thus transferring carriers in the high-energy states that contribute substantially to the SE

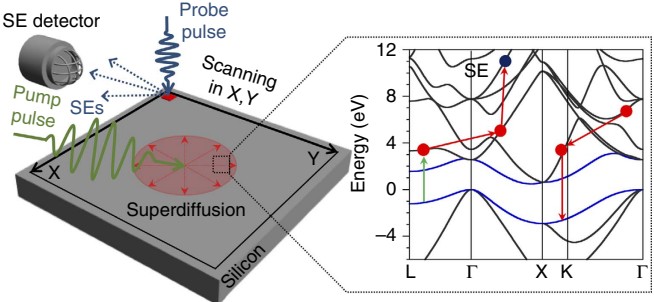

**Figure 1 | Schematic of scanning ultrafast electron microscopy (SUEM) imaging.** In our SUEM measurements, the optical pump pulse excites electron-hole pairs in silicon, which undergo super-diffusion until they equilibrate with the environment. Electron probe pulses with a known delay raster-scan the surface, resulting in the emission of secondary electrons (SEs) employed to image the carrier dynamics. The right panel schematically shows carrier dynamical processes mapped onto the bandstructure of Si. Shown are the carrier optical excitation (green arrow), as well as energy gain and recombination (upward and downward oriented red arrows, respectively) due to carrier scattering processes. The SEs preferentially observed by SUEM (blue dot) are emitted from the high-energy tail of the carrier distribution.

**a**

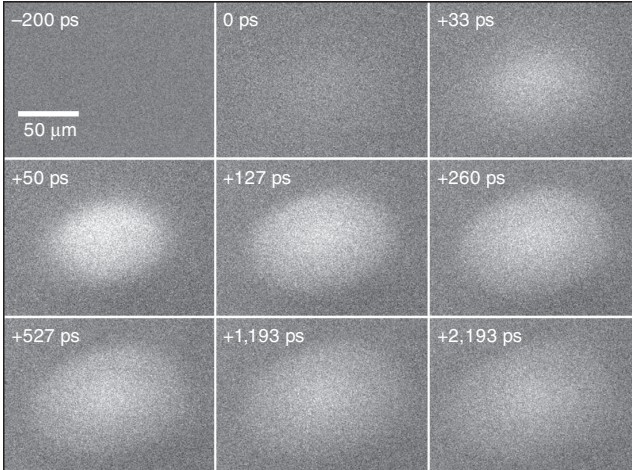

**b**

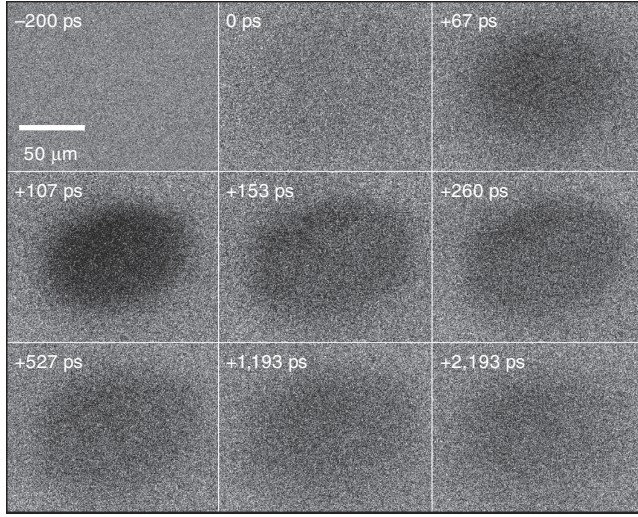

**Figure 2 | Electron and hole dynamics.** Scanning ultrafast electron microscopy (SUEM) images of the dynamics of (**a**) electrons and (**b**) holes in p-type and n-type silicon, respectively. The images show the unperturbed signal (at −200 ps) and the excitation (at time zero), ultrafast expansion (electrons: 33 − 260 ps; holes: 67 − 260 ps), and steady-state carrier diffusion (beyond 260 ps). Since the maximum brightness is achieved at 50 ps in **a** and 107 ps in **b**, the carrier diffusion is more clearly visible after these times. A comparison of the electron and hole dynamics at the same time delays is given in Supplementary Fig. 1.

emission signal (Fig. 1). The band bending and local potential at the surface can further delay SE emission[20].

At times greater than 50 ps, two main trends are visible in Fig. 2a. First, the electrons undergo cooling, thus decreasing their average energy and the SE signal intensity. Second, the electron population diffuses laterally, giving rise to broader spatial concentration profiles. The progressive decrease in brightness and increase in the size of the bright area are consistent with these two trends. Analysis of the population distributions (see below) shows that the lateral diffusion of the electrons is much faster at short times $t < 250$ ps than the typical diffusion in silicon at room temperature. For times longer than $\sim 500$ ps, a slower diffusion regime is established, with an electron diffusivity value typical of silicon at room temperature. The bright contrast ultimately disappears within 30 ns due to carrier recombination. We note

that no bright signal is visible at negative times, suggesting that the sample returns to equilibrium between two pulses separated by 200 ns. This guarantees that the measured spatiotemporal dynamics is not biased by the pulsed excitation.

Hole carriers in n-type silicon exhibit a similar spatiotemporal dynamics, as seen by the dark contrast images in Fig. 2b. Here the dark contrast is associated with a decreased electron concentration (with respect to the sample before illumination), representing the formation of an excess concentration of excited holes. The dark contrast reaches a maximum intensity at 107 ps, which is longer than the 50 ps for the the bright contrast maximum seen for electrons (Fig. 2a). We attribute this difference to the different initial energy distributions of the excited electrons and holes, and the fact that the electron mobility in silicon is three times higher than the hole mobility, which induces different carrier dynamics for electrons and holes near the surface.

Similar to the bright contrast for electrons, the hole spatial distribution becomes broader, expanding rapidly between 107 and 260 ps. In addition, the dark contrast decreases beyond 107 ps, due to cooling and lateral diffusion of the holes. Analysis of the hole spatial distributions (see below) shows that, similar to the electron case, after a transient characterized by very high diffusivity values, diffusion at $t > 260$ ps becomes significantly slower, reaching the room temperature diffusivity for holes in silicon at $\sim 500$ ps time delay. The sample then returns to equilibrium after 30 ns, as seen by the disappearance of the dark contrast. For completeness, contrast images showing the electron and hole distributions at the same set of delay times are given in Supplementary Fig. 1.

**Analysis of the carrier distributions**. To study the excited carrier transport quantitatively, we obtain the second moment of the electron and hole distributions, $\langle R^2 \rangle$, by separately analysing the contrast images of electrons and holes:

$$\langle R^2 \rangle(t) = \frac{\sum_{i,j} \left( x_i^2 + y_j^2 \right) I\left( x_i, y_j, t \right)}{\sum_{i,j} I\left( x_i, y_j, t \right)} \quad (1)$$

where $I(x_i, y_j, t)$ is the intensity of the pixel located at the discrete surface coordinates $(x_i, y_j)$ scanned by the electron probe, for the snapshot recorded at time $t$. The center of the distribution, which is determined from the data and does not shift during the dynamics, is taken to be the origin of the $x$ and $y$ axes in equation (1). In an ideally diffusive behaviour, $\langle R^2 \rangle$ is a linear function of time, with a slope proportional to the diffusivity $D$ (i.e., the diffusion coefficient). For the two-dimensional diffusion probed in the SUEM images, the second moment and diffusivity are related by $\langle R^2 \rangle = 4Dt$, so that the diffusivity is obtained as $D = (1/4) \cdot (\partial \langle R^2 \rangle / \partial t)$.

Figure 3 shows the time evolution of the second moment of the excited electron and hole distributions. We find two clearly distinct transport regimes for both electrons and holes. An initial transient diffusive regime is characterized by a large diffusivity $D \approx 10,000$ cm$^2$s$^{-1}$, namely, a value $\sim 100 − 1,000$ times higher than the room temperature diffusivity in silicon, $D_0 \approx 20 − 30$ cm$^2$s$^{-1}$ (ref. 18). This transient is associated with a super-diffusive expansion of the excited carrier spatial distribution, occurring on an ultrafast time scale of $100 − 200$ ps. A significantly slower diffusion regime rapidly sets in after $\sim 500$ ps, with associated diffusivity values (extracted from the second moment) of $30 − 40$ cm$^2$s$^{-1}$, in very good agreement with the room temperature diffusivity $D_0$[18]. While both electrons and holes exhibit the transient super-diffusive behaviour, the

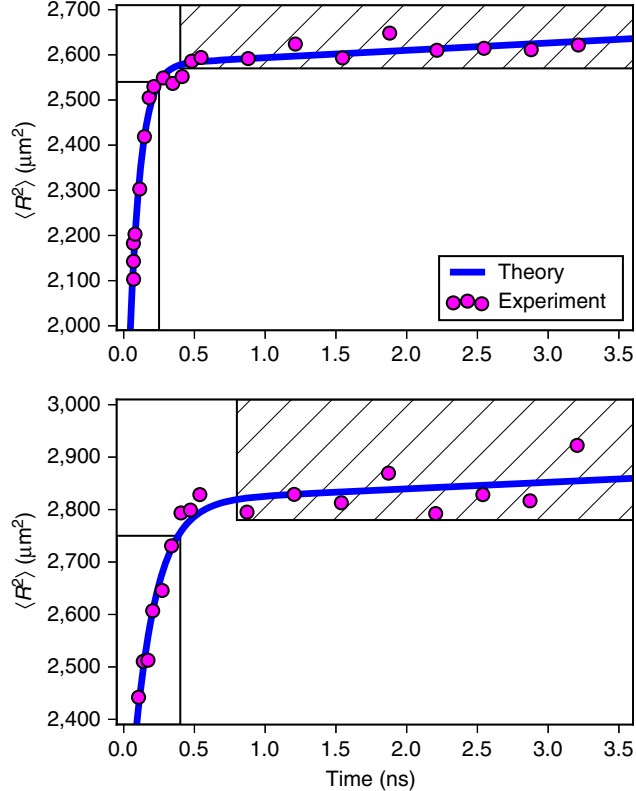

**Figure 3 | Second moment and the two diffusion regimes.** Time evolution of the second moment $\langle R^2 \rangle$ (equation (1)) of the spatial carrier distributions for excited electrons in p-type Si (top panel) and holes in n-type Si (bottom panel). The magenta circles are experimental data, and the blue line the numerical results obtained by solving the diffusion equation. The data in the boxes are used to obtain the transient diffusivity (empty box) and steady-state diffusivity (hatched box).

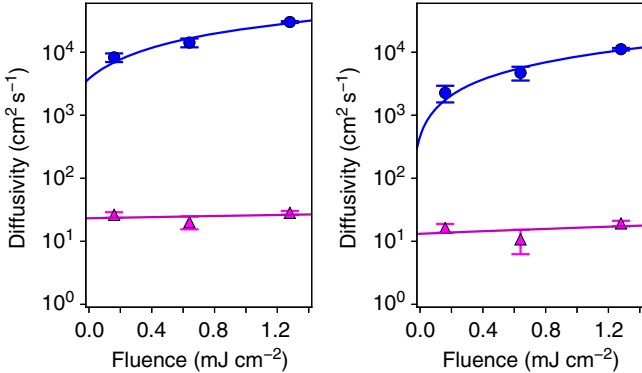

**Figure 4 | Laser fluence dependence.** Diffusivity as a function of laser fluence for electrons (left panel) and holes (right panel). The diffusivity values are given for the two different transport regimes found here, namely, ultrafast super diffusion (blue circles) and slow diffusion (magenta triangles). Linear fits of the data are also shown. The error bars are the s.d. obtained from the linear fit of $\langle R^2 \rangle$ versus time in the contrast images. The results are the average of three separate measurements.

transient diffusivity values of the electrons are higher by a factor of $2-3$ than those of the holes, consistent with the higher value of the electron mobility.

Figure 4 shows the diffusivity as a function of laser fluence in the two regimes found here, namely, the super-diffusive transport and the slower steady-state diffusive regime. The transient diffusivity is seen to increase monotonically with fluence for both electrons and holes. It follows a nearly linear trend, increasing from $\sim 8,000\,\text{cm}^2\,\text{s}^{-1}$ at $0.16\,\text{mJ}\,\text{cm}^{-2}$ fluence to $\sim 30,000\,\text{cm}^2\,\text{s}^{-1}$ at $1.28\,\text{mJ}\,\text{cm}^{-2}$ fluence for the electrons, and from $\sim 2,000$ to $\sim 11,000\,\text{cm}^2\,\text{s}^{-1}$ for the holes over the same fluence range. The slow diffusivity, on the other hand, remains relatively constant as a function of fluence. This trend is consistent with our interpretation of the slow diffusion regime as the steady-state diffusion of carriers in equilibrium with the environment, with an associated room temperature diffusivity $D_0$ given above.

**Theory and numerical modelling.** In what follows, we present a model to describe transport of excited carriers in our experiments. Establishing the initial conditions for the model is particularly important, as explained next. Using the experimental absorption in silicon at 515 nm wavelength[21], we estimate that excited carrier concentrations of $8 \times 10^{19}\,\text{cm}^{-3}$ are induced by the laser. For the laser power density used here, which is roughly $5\,\text{GW}\,\text{cm}^{-2}$ for the highest fluence of $1.28\,\text{mJ}\,\text{cm}^{-2}$, two-photon

absorption processes in Si[22] are negligible as they make up only 2% of the optical transitions. The average initial excited carrier temperature $T^*(0)$ can be estimated from the energy $E$ given out to the carriers by a photon[23], with $E = 2.4\,\text{eV}$ in our experiment. We thus obtain $T^*(0) = E/k_B \approx 28,000\,\text{K}$ in our conditions, namely, an initial carrier average temperature that far exceeds the Fermi temperature, leading to the formation of a non-degenerate carrier gas.

Following absorption, fast (tens of fs) electron-electron scattering processes rapidly establish a broad distribution of carrier energies; for carriers in the high-energy tail, the effective temperatures can be significantly greater than the average temperature. Gitomer et al.[24] have obtained the highest temperature reached by hot carriers under high-intensity laser illumination by measuring the bremsstrahlung x-ray spectrum. For the $\sim 5\,\text{GW}\,\text{cm}^{-2}$ power density employed here, they found the highest carrier energy to be of order 30 eV, that is, a maximum $T^*(0) = 3.5 \times 10^5\,\text{K}$. Because the SUEM technique measures emitted SEs, it is highly sensitive to the carrier energy, with more energetic electrons leading to higher SE emission. We reason therefore that our measurements, which rely on emitted SEs, preferentially observe the diffusion of carriers in the high energy tail of the excited carrier distribution (Fig. 1). Accordingly, we expect to observe effective carrier temperatures in the $\sim 1-5 \times 10^5\,\text{K}$ range. Given the large carrier kinetic energies associated with these temperatures, we neglect electron-electron interactions in our model.

In our non-degenerate excited carrier gas, the local equation of state can thus be approximated with that of an ideal gas, $P(\mathbf{r},t) = c(\mathbf{r},t)k_B T^*(\mathbf{r},t)$, where $P$ is the pressure, $c(\mathbf{r},t)$ the carrier concentration at position $\mathbf{r}$ and time $t$, $k_B$ is Boltzmann's constant, and $T^*(\mathbf{r},t)$ is the effective temperature of the excited carriers, which are assumed to achieve a hot Fermi-Dirac distribution shortly ($\sim 1\,\text{ps}$) after excitation[2]. Since the light absorption depth in silicon at 515 nm wavelength is orders of magnitude greater than the $\sim 10\,\text{nm}$ depth for SE emission, the excited carrier concentration is approximately constant in the surface-normal direction within 10 nm below the surface. We can thus use polar coordinates to describe the in-plane diffusion of the excited carriers, and further approximate the carrier concentration by removing its angular dependence (that is, assuming circular in-plane symmetry). The shape of the initial carrier concentration is chosen to be a gaussian in the

radial coordinate $r$, consistent with the SUEM image profile and reflecting the shape of the laser pulse. The carrier distribution at all times, normalized to the total excited carrier concentration $c_0$, reads:

$$c(r,t) = \frac{c_0}{\pi \sigma^2(t)} e^{-\frac{r^2}{\sigma^2(t)}} \qquad (2)$$

where $\sigma^2(t)$ is the second moment of the spatial concentration distribution, that is, the physical quantity $\langle R^2 \rangle$ used above to analyse the SUEM images (equation (1)). The gaussian laser beam used in the experiment is elliptical with 40 and 50 μm minor and major radii, respectively. We approximate it with a radial gaussian beam, and fit its initial width to experiment.

The experimentally observed linear dependence of $\langle R^2 \rangle$ versus time in both the transient and steady state regimes inherently implies a diffusive regime for carrier transport. We thus derive a diffusion model to explain the observed experimental trends. The diffusion equation employed in this work can be obtained from the continuity equation, $\partial c / \partial t = - \nabla \cdot \mathbf{J}$, where $\mathbf{J}$ is the carrier flux. In the typical treatment of carrier diffusion at room temperature in semiconductors[25], the flux is proportional to the concentration gradient, $\mathbf{J} = - D_0 \nabla \mathbf{c}$, leading to the diffusion equation $\partial c / \partial t = D_0 \nabla^2 c$. Here we generalize this approach by treating carrier transport as the result of the large pressure gradient within the excited carrier gas. We write the flux as $\mathbf{J} = c(r,t)\mathbf{v}$, where the velocity $\mathbf{v} = \mu \mathbf{F}/e$ is defined by the force $\mathbf{F} = - \nabla \mathbf{P}/c$ due to the pressure gradient; $\mu$ is the electron mobility and $e$ the electron charge. Using the equation of state, we find:

$$\mathbf{J}(r,t) = c(r,t)\mathbf{v} = \frac{\mu k_B}{e}(c\nabla \mathbf{T}^* + T^* \nabla \mathbf{c}) \qquad (3)$$

We neglect the temperature gradient within the area excited by the laser, and assume that the effective temperature of the excited carriers decays exponentially with a characteristic relaxation time $\tau$:

$$T^*(t) = T^*(0)e^{-t/\tau} + T_0 \qquad (4)$$

to the equilibrium environment temperature $T_0$ (here, 300 K), starting from an initial excited carrier temperature $T^*(0) \gg T_0$. Substituting $T^*(t)$ in equation (3), neglecting the temperature gradients and using the continuity equation, we obtain a modified diffusion equation in the radial coordinate $r$:

$$\frac{\partial c(r,t)}{\partial t} = [D_0 + D^*(t)]\nabla^2 c(r,t) \qquad (5)$$

where $D_0 = \mu k_B T_0 / e$ is the Einstein relation[25] for the room temperature diffusivity of thermalized carriers, and $D^*(t) = [\mu k_B T^*(0)e^{-t/\tau}]/e$ is the transient diffusivity due to the excess energy of the excited carriers. It is seen that $D^*(t)$ decays exponentially over a time scale of a few relaxation times. Note that we neglect the temperature dependence of the mobility, which is weak for our highly doped samples[26] and controlled by the lattice (as opposed to the much higher electronic) temperature. We stress that our model, through equation (4), implies a continuous cooling of the carriers, with a relaxation time dependent on the particular experimental conditions.

We solved equation (5) both analytically and numerically with a zero-flux boundary condition at large $r$ (see the Methods section). Given the initial gaussian distribution, it can be shown straightforwardly that equation (5) yields a gaussian distribution at all times, so that the spatial dynamics can be characterized by the second moment $\langle R^2 \rangle(t) = \sigma^2(t)$, computed here from the simulations. Figure 3 shows our numerical results for the second moments of electron and holes as a function of time, along with the experimentally obtained second moments. Our results correctly predict the presence of two distinct diffusion regimes (in spite of the assumed continuous cooling), namely, transient and steady-state diffusion in the presence of an exponentially decaying diffusivity. Our simulations show that the carrier spatial distribution expands rapidly due to the large concentration and pressure gradients. Once the transient diffusivity term $D^*$ dies out after $\sim 500$ ps, a steady state diffusion regime with diffusivity $D_0$ is reached. When relaxation times and initial conditions obtained from experimental fits are employed, our model achieves excellent quantitative agreement with the experimentally measured second moments, in both the transient and steady state regimes (see Fig. 3). To obtain the numerical data in Fig. 3, we employ best-fit values for the relaxation time $\tau$ and initial temperature $T^*(0)$ of $\tau = 77$ ps and $T^*(0) \approx 4 \times 10^5$ K for the electrons, and $\tau = 161$ ps and $T^*(0) \approx 2.7 \times 10^5$ K for the holes. As discussed above, these effective carrier temperatures of order $1 - 5 \times 10^5$ K are consistent with our experimental conditions, and are a consequence of the fact that SUEM preferentially measures the signal from high-energy carriers within the excited carrier distribution. We believe other carriers with lower temperature and diffusivity, associated with weaker signal than the higher energy fast-diffusing carriers, are also present though not directly observed.

The relaxation time can be interpreted as the decay time for the sample to return to equilibrium with the environment after excitation with the laser pulse. In this process, the carriers lose energy through phonon emission with a time scale of $\sim 1$ ps eV$^{-1}$ (ref. 10), giving an overall cooling time of up to $10 - 50$ ps for the initial carrier energies in our experiment. Note that the high carrier density regime probed here is distinct from the low density regime, in which hot carriers thermalize in a few ps[10,27]. Different from the low density regime, both hot phonon effects, which are known to increase the carrier cooling times by 2–3 orders of magnitude at our carrier density, and screening of the electron-phonon interaction can contribute in our experiment to increasing the diffusivity decay time constant. While the relaxation time generally depends on carrier density, this dependence does not affect our measured transient diffusivity, since for each given fluence value the average carrier density is nearly constant throughout our experiment. The phonon population also equilibrates and dissipates heat to the environment, with timescales typically one order of magnitude longer than electron cooling. Overall, the known timescales for electron and phonon dynamics are consistent with the $100 - 500$ ps equilibration times obtained here.

**First-principles calculations**. We close this section by presenting first-principles density functional theory (DFT) calculations (see the Methods section). To obtain microscopic insight into the diffusion dynamics, we treat the diffusion process as a two-dimensional random walk of excited carriers. Since the pump pulse is short and does not overlap in time with the diffusive dynamics, on the time scale of interest (10 ps − 10 ns) the carriers move randomly at their average thermal velocity, and scatter with ionized impurities and phonons with an average scattering time $\tau_S$. For carriers with an average square thermal velocity $\langle v^2 \rangle$, the diffusivity can be written as $D = \langle v^2 \rangle \tau_S / 4$ (ref. 28). We compute the bandstructure and the density of states of silicon to obtain the average square thermal velocity $\langle v^2 \rangle(T)$ as a function of electron temperature $T$, for $T$ up to $\sim 10^6$ K (see the Methods section). We use this information together with the experimental room temperature diffusivity, $D_0 \approx 30$ cm$^2$ s$^{-1}$, to estimate the scattering time $\tau_S = 4D_0/\langle v^2 \rangle(T)$ for electrons at room temperature, and obtain $\tau_S \approx 0.8$ ps. The Brooks–Herring formula[26] for ionized impurity scattering, applied to silicon in the

same conditions, yields a similar value of $\tau_S \approx 0.4$ ps, in agreement with the fact that at room temperature electron scattering in silicon with a dopant concentration of $10^{19}$ cm$^{-3}$ is dominated by ionized impurity scattering[26].

We then estimate the initial excited electron temperature $T^*(0)$ using the following argument. As the electrons cool after excitation with the pump pulse, the lattice temperature increases as energy is transferred to the lattice through electron-phonon coupling. While the electron temperature reaches high values of up to $\sim 10^5$ K, the lattice temperature increase is limited by the much higher heat capacity of the lattice compared to the carriers, and also by the fact that the sample dissipates heat into the environment. From the fact that the samples do not melt on the surface, we infer the lattice temperature never exceeds $\sim 1,000$ K in our experiments. The electron mobility, which is proportional to the scattering time $\tau_S$, is controlled by electron-phonon scattering (and thus by the lattice temperature) above $\sim 500$ K for our dopant concentration[26]. The mobility varies only by less than an order of magnitude in the $300 - 1,000$ K range for our dopant concentration[26], an indication that $\tau_S$ is only moderately affected by the increased lattice temperature. For these reasons, we assume that $\tau_S$ is nearly unchanged after the pump excites the carriers.

We thus use the scattering time $\tau_S$ obtained at room temperature, together with $\langle v^2 \rangle (T)$ obtained from DFT, to compute the initial excited electron temperature $T^*(0)$ for which $D^* = \langle v^2 \rangle (T^*(0)) \tau_S / 4$, where $D^* \approx 30,000$ cm$^2$ s$^{-1}$ is the experimental electron diffusivity in the super-diffusive regime. This calculation yields an estimated value of the initial electron temperature of $T^*(0) \approx 8 \times 10^5$ K, in very good agreement with our fitted electronic temperature of $4 \times 10^5$ K. This result confirms that the super-diffusive regime is consistent with the random thermal motion of a very hot degenerate electron gas generated at the surface by the pump pulse.

## Discussion

The optimal fitting parameters employed in our diffusion model lead to carrier dynamics in excellent agreement with experiment. We quantitatively reproduce the observed experimental trends, and explain the origin of the ultrafast super-diffusion in terms of a large transient diffusivity induced by the excess energy of the excited carriers. This picture is confirmed by our first-principles calculations. We highlight that our model does not bias the interpretation of the experimental data; it merely relies on diffusion theory, in accordance with the observed trends for the second moments. While in this work we employ fitting parameters and simple first-principles calculations, future work will attempt to model SUEM experiments entirely from first principles[10,11] by studying SE emission in the presence of a time-dependent excited carrier population.

In summary, we characterize the spatiotemporal dynamics of excited carriers in silicon, and for the first time directly observe a super-diffusive transport regime originating from the high effective temperature and kinetic energy of a fraction of the excited carriers. These findings are critical for understanding ultrafast charge and spin dynamics in materials.

## Methods

**Experimental details.** We employ p-type and n-type silicon wafers with doping concentrations of $2.1 \times 10^{19}$ cm$^{-3}$ and $1.2 \times 10^{19}$ cm$^{-3}$, respectively. They were purchased from MTI Corp and used without modifications. The samples are cleaved and immediately transferred into the high vacuum ($1.2 \times 10^{-6}$ Torr) SUEM chamber. To obtain spatiotemporal evolution images with SUEM (see below), the samples were excited with a 515 nm wavelength gaussian laser pulse with a fluence in the $0.16 - 1.28$ mJ cm$^{-2}$ range and a 5 MHz repetition rate. The SUEM operation is briefly summarized here, and discussed extensively elsewhere[29]. In the SUEM, femtosecond infrared pulses (1,030 nm wavelength, 300 fs duration, 5 MHz repetition rate) were generated from a Clark-MXR

fiber laser system and used to produce green (515 nm) and UV (257 nm) pulses. The green light excites the specimen while the UV light is directed toward the photocathode, which in response emits short electron pulses employed to probe the carrier dynamics (Fig. 1). The time delay between the pump and probe pulses is controlled by a delay line, covering a range between $-760$ ps to 3.32 ns, here with a snapshot resolution of 2.0 ps. The electron pulses, accelerated to 30.0 kV by electrostatic lenses, raster-scan the surface, generating secondary electrons (SEs) which are then collected by an Everhart–Thornley detector and translated into pixel intensities to form images. Our measurements achieve a 200 nm resolution sufficient to resolve the μm-scale spatial dynamics examined in this work. We have additionally shown in previous work[29] that this resolution can be made better than 10 nm, enabling studies of carrier dynamics in nanomaterials. While the volume excited by the electron pulse can extend deep into the sample, the SEs are only emitted from the top $5 - 10$ nm below the surface, thus providing surface sensitive information[30].

**Numerical solution of the diffusion equation.** We implement and solve the diffusion equation using MATLAB. We employ polar coordinates, and use an adaptive time step integrator to evolve in time the carrier concentration, which is approximated at time zero by a gaussian in the radial coordinate. The second moment of the initial distribution is obtained by fitting the images with maximum contrast, which are collected at delay times of 50 ps for electrons and 107 ps for holes. These times are chosen, respectively, as the initial times for the numerical simulations. Note that our model for carrier diffusion can also be solved analytically; the numerical simulations are employed both to validate the analytical solution and to investigate the effect of a non-uniform temperature, which cannot be studied analytically, as described next. The extent of the spatial region over which the excited carriers thermalize is not obvious a priori. Several scattering events are needed to establish a hot Fermi-Dirac distribution, and thus a region spanning several mean free paths is necessary to establish a local temperature within the carrier gas. Given the large thermal velocities of the excited carriers, this region can span the entire laser spot area. Accordingly, the model presented in the manuscript assumes a nearly constant electron temperature in the region of interest for the dynamics, which allows us to neglect the temperature gradient in equation (3). We have also carried out numerical simulations for a model that includes a spatially varying temperature, which is included in the diffusion equation through the temperature gradient in equation (3). We approximate the initial temperature distribution with a gaussian spatial profile with the same shape as the intensity profile of the laser. Using this non-uniform temperature distribution, which is taken to decay exponentially in time as for the uniform distribution in equation (4), we obtain a diffusion equation that cannot be solved analytically. We solve this modified diffusion equation by time-stepping both the carrier concentration and their temperature (see Supplementary Fig. 2 for details), and find that the second moment $\langle R^2 \rangle (t)$ and the carrier concentration are nearly the same as in the simplified model with uniform temperature presented above. The results comparing the second moment $\langle R^2 \rangle (t)$ for the uniform and gaussian spatial temperature distributions are shown in Supplementary Fig. 2.

**First-principles calculations.** We carry out density functional theory (DFT) calculations on silicon with a relaxed lattice constant of 10.2 bohr, using the local density approximation (LDA)[31] and a plane wave basis with the Quantum ESPRESSO code[32]. Norm-conserving pseudopotentials[33] and a plane-wave kinetic energy cutoff of 90 Ry are employed to obtain the ground state charge density. The density of states $D(E)$ is obtained for energies $E$ up to 500 eV above the conduction band minimum (CBM) with a $100 \times 100 \times 100$ **k**-point grid in the Brillouin zone (BZ). The bandstructure in the [100] direction, which is taken here to be the x direction and corresponds to the $\Gamma - X$ direction in the BZ, is also computed for energies up to $\sim 500$ eV above the CBM. The calculation includes $\sim 1,000$ empty bands and a fine **k**-point grid in the $\Gamma - X$ direction. The square velocity in the x direction, $(v_{n\mathbf{k}}^2)_x$, is computed for each Kohn-Sham state with band index $n$ and crystal momentum **k** as $(v_{n\mathbf{k}}^2)_x = [(1/\hbar) \cdot (\partial E_{n\mathbf{k}} / \partial \mathbf{k})]^2$, using a discrete derivative obtained from the energy of adjacent **k**-points along $\Gamma - X$. The square velocities are then binned in energy and averaged within the energy bin, to obtain an average square velocity $\langle v^2 \rangle (E)$ as a smooth function of the electron energy $E$. This quantity is then employed, together with the density of states, to compute the average square velocity for the electrons as a function of temperature:

$$\langle v_x^2 \rangle (T) = \frac{\int_{E_C}^{\infty} dE [D(E) \cdot f(E, T, \mu)] \cdot \langle v^2 \rangle (E)}{\int_{E_C}^{\infty} dE [D(E) \cdot f(E, T, \mu)]} \quad (6)$$

where $T$ is the electronic temperature, and $f(E, T, \mu)$ the Fermi-Dirac occupation for a chemical potential $\mu$, which for our value of $\sim 10^{19}$ cm$^{-3}$ excited electrons is $\sim 50$ meV below the CBM energy $E_C$[26].

Two sets of calculations are then carried out. At room temperature (here $T = 300$ K), we sample the CBM with a fine **k**-point grid, so that the $\sim 25$ meV energy window above the CBM for which $f(E, T, \mu)$ is non-zero is accurately sampled. Using the square velocity at 300 K computed using equation (6), we first compute the room temperature carrier scattering time $\tau_S$ from the experimental

value of the diffusivity $D_0$ at room temperature:

$$\tau_S = \frac{2 \cdot D_0}{\langle v_x^2 \rangle (T = 300\,\text{K})} \qquad (7)$$

where we used the equation given in the main text, $\tau_S = 4D_0/\langle v^2 \rangle (T)$, together with $\langle v_x^2 \rangle = \langle v^2 \rangle / 2$, as is appropriate for a two-dimensional isotropic surface. Using the scattering time computed at room temperature, and thus neglecting the change in the scattering time on excitation as discussed in the main text, we compute the initial excited carrier temperature $T^*(0)$ in the super-diffusive regime by solving the equation:

$$D^* = \langle v_x^2 \rangle (T^*(0)) \tau_S / 2 \qquad (8)$$

where $D^* \approx 30,000\,\text{cm}^2\,\text{s}^{-1}$ is the electron diffusivity in the super-diffusive regime. We obtain a value of $T^*(0) = 8 \times 10^5\,\text{K}$, in very good agreement with our fitted electronic temperature of $4 \times 10^5\,\text{K}$.

**Data availability.** The data that support the findings of this study are available from the corresponding author on request.

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

## Acknowledgements

This work was supported by NSF grant DMR-0964886 and Air Force Office of Scientific Research grant FA9550-11-1-0055 in the Physical Biology Center for Ultrafast Science and Technology at California Institute of Technology, which is supported by the Gordon and Betty Moore Foundation. M.B. thanks the California Institute of Technology for a start-up fund. V.I. and M.B. gratefully acknowledge support by the Caltech-GIST program. This research used resources of the National Energy Research Scientific Computing Center, a DOE Office of Science User Facility supported by the Office of Science of the U.S. Department of Energy under Contract No. DE-AC02-05CH11231.

## Author contributions

E.N. and A.Z. conceived and designed the experiments. E.N. carried out the experiments and analysed the data. V.I. and M.B. developed the theory and carried out numerical simulations and *ab initio* calculations. E.N. and M.B. wrote the manuscript. All authors reviewed the manuscript.

## Additional information

**Competing interests:** The authors declare no competing financial interests.

**Publisher's note**: 

