## [Peer Review File · Nature Communications]

Reviewers' comments:

Reviewer #1 (Remarks to the Author):

The main issue of this paper is the super diffusion of the photoexcited carriers in the doped Si, revealed with the scanning ultrafast electron microscopy. They evaluated the spatiotemporal dynamics of the excited carrier and found the huge diffusion coefficient compared with that at the low-density regime. I'm not surprised at these results. According to the paper, Young & Van Driel, PRB 36, 2147 (1982), diffusive coefficient strongly depends on the carrier density due to the many-body effects. It is enhanced above 10^{19} cm⁻³, which is the same density regime as the present experiments (10^{19} - 10^{20} cm⁻³). There are many papers related to it in the viewpoints of highly-excited semiconductor, including laser induced melting of the semiconductors, and the some experimental reports support it. Therefore, present results should be treated as "dense carrier diffusion" rather than "hot carrier diffusion". In this case, the authors have to investigate the diffusion coefficient in the low density regime carefully, and have to discuss anomalous diffusion in the viewpoint of the many-body effects. I comment that this paper is not suitable for the publication in Nature Communication.

Reviewer #2 (Remarks to the Author):

The present work provides a novel approach to monitor and unravel the short time dynamics of electron hole pairs created after a short laser excitation in silicon. Using ultrafast scanning tunneling microscopy determined, by measuring the second moment of the electron and hole distributions, two regimes for the spatiotemporal evolution of the electron (hole) populations after laser excitation (with a time and space resolutions are, respectively, 2 ps and 200 nm). The main results is the observation of a transient super-diffusive behavior below 200ps (nearly three orders of magnitude larger than the room T) . A very simple model that treats the excited carriers as a hot non-degenerate electron gas is able to capture all the main effects in the experiment. In particular it allow the authors to attribute the rapid transient expansion to the large pressure gradient in the carrier population induced by the laser pulse. I think the paper is sound and should be published, however I have some questions that really need to be addressed before the paper can be consider for publication:

- Can the authors estimate the local heating of the sample due to the strong applied laser. Could some of the observed effects be linked to a high T gradient in the electron/hole distribution compared to the lattice.
- The effect discussed here should be general and should be also observable if the pump laser is long (monochromatic) and we problem the electron distribution while the pump is acting. Is this correct? Could the authors discuss the impact of the temporal shape of the applied laser.
- What happen if the experiments are done on weakly doped Si samples? Is there any connection between the super diffusive behavior observed here and the fact that the samples are heavily p or n-doped?
- Concerning the modeling part, I am a little bit surprised that such a simple model based on a hot non-degenerate electron gas works. I would have expected a rather inhomogeneous electron/hole population (used on the fact that only e-h pairs along specify directions are optical excited). Ab initio calculations should shed some light into the process (one of the authors has done amor contributions to describe the carrier lifetime in doped semiconductors). I would suggest the authors to include some abinitio modeling supporting their model. This would really make the paper more fundamentally ground.

Jus a minor point, I am not convinced about the claim that "The findings open new avenues for the potential control of ultrafast spatial dynamics of excited carriers in materials". This works provides evidence of an ultrafast diffusive mechanism abut not any proposal on how to control it.

Reviewer #3 (Remarks to the Author):

The manuscript by Najafi et al reports experiments in which a silicon surface is optically excited with a Gaussian laser pulse, and the spatio-temporal dynamics of the excited charge carriers subsequently followed by raster-scanning the surface with a focussed electron beam and measuring the secondary electron emission. The expansion of the spatial distribution of excited carriers is used to identify a cross-over from initial super-diffusive transport driven by the high pressure gradient accompanying the creation of a locally hot electron gas, to normal room-temperature carrier diffusion after 500ps. The interpretation is supported by a theoretical model and numerical calculation, which give convincing support to the interpretation.

I find the work novel and worthy of publication in Nature Communications. The ability to directly observe transient dynamics, and particularly super-carrier diffusion is extremely interesting and notable, and offers the potential for new insight which can impact upon a range of physics involving ultrafast spatial dynamics.

There are some matters which should be considered before final acceptance, listed below, but once dealt with the work deserves publication.

- * The authors should say more about why the SE intensity can be taken as directly proportional to excited carrier concentration.
- * Although the authors also include two movies of the evolution of the SUEM images, I consider Fig. 2 would be improved if similar frame times were included for both n- and p-type samples. i.e. p-type should include 67, 107 and 153 ps frames, and n-type 33, 50 and 127 ps. This would better convey the differences between the evolutions in the article itself.
- * The calculation of the second moment via Eqn. (1) should be clarified. This assumes the center of the SE distribution was at $(x,y)=(0,0)$, in conflict with the impression given by Fig. 1. Were $\langle X \rangle$, $\langle Y \rangle$ determined from the data?
- * p5 para 2. $\langle R \rangle^2$ should be $\langle R^2 \rangle$.
- * Eqn. (2) is not an expression for the initial distribution, as stated, but actually describes a time-dependent concentration with Gaussian profile and time-dependent variance. The presentation of this form precedes the discussion of the model which goes on to predict this temporal evolution. Eqn (2) should be an expression for $c(r,0)$ with σ^2 defined as $\langle R^2 \rangle(0)$.
- * I am a little unclear regarding the role of the numerical modelling, stated as being used to solve Eqn. (5). This equation has an analytic solution, as mentioned in the text: $c(r,t)$ is given by Eqn (2), with $\sigma^2(t)$ also obtainable analytically given the assumed form for $D^*(t)$. Or am I missing something?
- * Why is it appropriate to use the same mobility in the room temperature diffusivity, and that of the excited carriers?

REVIEWER REPORTS

REFEREE 1

Comment:

The main issue of this paper is the super diffusion of the photoexcited carriers in the doped Si, revealed with the scanning ultrafast electron microscopy. They evaluated the spatiotemporal dynamics of the excited carrier and found the huge diffusion coefficient compared with that at the low-density regime. I'm not surprised at these results. According to the paper, Young & Van Driel, PRB 36, 2147 (1982), diffusive coefficient strongly depends on the carrier density due to the many-body effects. It is enhanced above 10^{19} cm^{-3} , which is the same density regime as the present experiments (10^{19} - 10^{20} cm^{-3}). There are many papers related to it in the viewpoints of highly-excited semiconductor, including laser induced melting of the semiconductors, and some experimental reports support it. Therefore, present results should be treated as "dense carrier diffusion" rather than "hot carrier diffusion". In this case, the authors have to investigate the diffusion coefficient in the low density regime carefully, and have to discuss anomalous diffusion in the viewpoint of the many-body effects. I comment that this paper is not suitable for the publication in Nature Communication.

Response to comment: We thank the referee for raising a concern about our work. We respectfully disagree with the comment made by the referee that the observed super-diffusion is merely due to the carrier density, for the following reasons:

- 1) The super-diffusion observed in our experiment lasts for only ~ 200 ps. After this transient, the number of carriers is nearly unchanged, yet the diffusivity at steady state rapidly drops to typical room temperature values, which are $\sim 10,000$ times smaller than those observed in the super-diffusion transient regime. Because the carrier density is nearly constant throughout the measurement, while the diffusivity changes by 4 orders of magnitude, it is clear that super-diffusion cannot be a mere consequence of the carrier density. The same is true for many-body effects due to the carrier density: while these are present both in the transient and at steady state, the diffusivity differs by 4 orders of magnitude in these two states. In our experiment, carrier super-diffusion is evidently related to a transient dynamics induced by the laser pulse excitation.
- 2) The paper mentioned by the referee (Young et al., PRB 26, 2147, 1982) compares two different theories to compute the diffusivity at 300 K as a function of carrier density. The two theories give diffusivities that agree within a factor of 2 of each other. In our work, the excess energy transferred by the laser pulse induces electronic temperatures of tens of thousands of degrees, and results in 4 orders of magnitude higher diffusivities than those of carriers at room temperature. Therefore, our conditions are entirely different from those analyzed in the work mentioned by the referee.
- 3) It is puzzling that the referee suggests an explanation based on carrier density alone: large carrier densities can be realized routinely in MOSFET transistors and quantum wells, yet the carriers clearly do not super-diffuse in those devices.

Our treatment of excited carrier dynamics is consistent with the literature and with physical intuition. Two referees agree with us that the explanation of our experimental results is sound and well reasoned.

REFEREE 2

Introductory comment:

The present work provides a novel approach to monitor and unravels the short time dynamics of electron hole pairs created after a short laser excitation in silicon. Using ultrafast scanning tunneling microscopy determined, by measuring the second moment of the electron and hole distributions, two regimes for the spatiotemporal evolution of the electron (hole) populations after laser excitation (with a time and space resolutions are, respectively, 2 ps and 200 nm). The main result is the observation of a transient super-diffusive behavior below 200ps (nearly three orders of magnitude larger than the room T). A very simple model that treats the excited carriers as a hot non-degenerate electron gas is able to capture all the main effects in the experiment. In particular, it allows the authors to attribute the rapid transient expansion to the large pressure gradient in the carrier population induced by the laser pulse. I think the paper is sound and should be published.

Response to comment: We thank the referee for the positive feedback, and for recommending publication of our manuscript in Nature Communications. We have revised our manuscript to fully address the referee's comments and suggestions. In particular, we have included ab initio calculations to estimate the initial temperature of the electron gas in the super-diffusive regime.

Comment #1:

Can the authors estimate the local heating of the sample due to the strong applied laser. Could some of the observed effects be linked to a high T gradient in the electron/hole distribution compared to the lattice.

Response to comment #1: We thank the referee for pointing out the effect of lattice temperature on the observed dynamics. The samples do not melt on the surface during the experiment, which indicates that the lattice temperature never exceeds $\sim 1,500$ K. While the electron temperature reaches much higher values of up to $\sim 10^5$ K, the lattice temperature increase is limited by the much higher heat capacity of the lattice compared to the carriers, and also by the fact that the sample dissipates heat into the environment.

Energy transfer between the excited electrons and the lattice is regulated by electron-phonon coupling, and is expected to occur on a time scale of ~ 10 ps. The phonons then thermalize and dissipate heat over a ~ 100 ps time scale, consistent with the time the carriers take to equilibrate and exhibit the steady-state diffusivity of ~ 30 cm²/s typical of Si at room temperature.

The spatial dependence of the temperature within the electron gas is also an interesting question. The region in space over which the carriers equilibrate is not obvious a priori; since several scattering events are needed to establish a hot Fermi-Dirac distribution, a length of a several mean free paths is necessary to establish a local temperature. Given the large thermal velocities of the excited carriers, this region can span the entire laser spot area. Accordingly, the model presented in the manuscript assumes a nearly constant electron temperature in the region of interest for the dynamics, which allows us to neglect the temperature gradient in eq. 3. We have also carried out numerical simulations for a model that includes a spatially varying temperature, which enters the diffusion equation through the temperature gradient in eq. 3. We approximate the initial temperature

distribution with a Gaussian spatial profile with the same shape as the intensity profile of the laser. Using this non-uniform temperature distribution, which is taken to decay exponentially in time as for the uniform distribution in eq. 4, we obtain a diffusion equation that cannot be solved analytically. We solve this modified diffusion equation by time-stepping both the carrier concentration and the temperature, and find that the second moment and the carrier concentration are nearly the same as in the simplified model with uniform temperature presented in the manuscript. The simulations comparing the second moment for the uniform and Gaussian spatial temperature distributions have been included in Figure S2 of the revised Supplementary Information.

Changes made in response to comment #1: We included a numerical simulation for a spatially varying temperature with a Gaussian profile. Figure S2 in the revised Supplementary Information compares the second moment obtained for the constant and Gaussian temperature distributions. The discussion above has been added on page 7, right column, of the revised manuscript.

Comment #2:

The effect discussed here should be general and should be also observable if the pump laser is long (monochromatic) and we problem the electron distribution while the pump is acting. Is this correct? Could the authors discuss the impact of the temporal shape of the applied laser.

Response to comment #2: We agree with the referee that a relatively similar dynamics should be observed when the sample is excited with a longer laser pulse. The duration of the laser pulse employed here is ~ 1 ps, and thus much shorter than the dynamics of interest. Repeating the experiment using nanosecond pulses is possible, and it is in fact part of our future plans. We expect that the main trends would be unchanged, and that the super-diffusing regime would last longer, at least ~ 200 ps after the end of the laser pulse, similar to the result found here. Using a continuous-wave laser to excite the sample, on the other hand, would make the experiment challenging: the spatiotemporal dynamics would average out in the signal, and it would likely not be revealed in the contrast images.

Comment #3:

What happen if the experiments are done on weakly doped Si samples? Is there any connection between the super diffusive behavior observed here and the fact that the samples are heavily p or n-doped?

Response to comment #3: We thank the referee for this comment. We observed similar super-diffusive dynamics in lightly and moderately doped silicon. Highly doped samples were chosen in this work to increase the signal-to-contrast ratio in the SUEM images in Fig. 2. The reason why the signal-to-contrast ratio improves for increasing doping is that the contrast images in SUEM are obtained by subtracting a ground state image of the sample, which is recorded in the dark before the pump pulse excitation. When a highly doped *p*-type sample is employed, the electron concentration in the ground state is small, so that the pump pulse induces a large excess of electrons relative to the ground state. In intrinsic or lightly doped samples, on the other hand, the same pump pulse would induce a smaller electron excess concentration relative to the ground state. Employing highly doped samples thus allows us to obtain images with high signal-to-contrast ratios with

reasonable values of the laser fluence. Significantly higher pump fluences are needed to observe strong signals in intrinsic and lightly doped silicon, and this often results in surface melting of the sample and other undesirable effects.

Comment #4:

Concerning the modeling part, I am a little bit surprised that such a simple model based on a hot non-degenerate electron gas works. I would have expected a rather inhomogeneous electron/hole population (used on the fact that only e-h pairs along specific directions are optical excited). Ab initio calculations should shed some light into the process (one of the authors has done more contributions to describe the carrier lifetime in doped semiconductors). I would suggest the authors to include some ab initio modeling supporting their model. This would really make the paper more fundamentally ground.

Response to comment #4: We agree that ab initio calculations can provide valuable microscopic insight into carrier dynamics. As we state on page 1, SUEM experiments are particularly challenging to describe from first principles. The experiments reported in this work would require modeling the spatial dynamics of carriers on the μm length and ns time scales; no known first-principles technique can achieve this result at present. For this reason, we employed a hydrodynamic model of the electron gas to qualitatively and semi-quantitatively explain the experimental results. We have long-term plans to develop a fully first principles description of SUEM.

We also agree with the referee that presenting at least some ab initio calculations would make the paper more complete. *We have included ab initio calculations in the revised manuscript.* We employed density functional theory to compute the bandstructure and density of states in silicon, and employed this data to obtain the average thermal velocity of the electrons and their scattering time, as well as the initial temperature of the excited carriers in the super-diffusive regime, which is in very good agreement with our fitted electronic temperature. These calculations are presented on page 6 of the revised manuscript, and detailed in the Methods section. We thank the referee for this constructive criticism, which has helped us improve our manuscript. We believe that the ab initio results, while simple, make the modeling part of the manuscript stronger.

Changes made in response to comment #4: We have carried out ab initio calculations, and included them in the revised manuscript on page 6 and in the Methods section.

Comment #5:

Jus a minor point, I am not convinced about the claim that “The findings open new avenues for the potential control of ultrafast spatial dynamics of excited carriers in materials”. This works provides evidence of an ultrafast diffusive mechanism about not any proposal on how to control it.

Response to comment #5: We are grateful for this comment, and agree with the referee that improved carrier control, which we claim in the abstract, is not shown in our work. We envision that one could employ super-diffusion to manipulate carriers in ways different than what is possible with conventional room temperature diffusion. Since it is not our intention to exaggerate the implications of our work, we have changed the sentence in the abstract to reflect this helpful criticism.

Changes made in response to comment #5: We changed the last sentence in the abstract to: “Our findings open new avenues for investigating ultrafast spatial dynamics of excited carriers in materials.”

REFeree 3

Introductory comment:

The manuscript by Najafi et al reports experiments in which a silicon surface is optically excited with a Gaussian laser pulse, and the spatio-temporal dynamics of the excited charge carriers subsequently followed by raster-scanning the surface with a focused electron beam and measuring the secondary electron emission. The expansion of the spatial distribution of excited carriers is used to identify a cross-over from initial super-diffusive transport driven by the high pressure gradient accompanying the creation of a locally hot electron gas, to normal room-temperature carrier diffusion after 500ps. The interpretation is supported by a theoretical model and numerical calculation, which give convincing support to the interpretation.

I find the work novel and worthy of publication in Nature Communications. The ability to directly observe transient dynamics, and particularly super-carrier diffusion is extremely interesting and notable, and offers the potential for new insight which can impact upon a range of physics involving ultrafast spatial dynamics.

Response to comment: We are grateful for the very positive comments, and also for recommending our work for publication in Nature Communication. We are delighted that the referee shares our excitement about novel techniques for studying the ultrafast spatial dynamics of excited carriers. We have fully taken into account the constructive criticism by the referee, which we address in this letter and in the revised manuscript.

Comment #1:

The authors should say more about why the SE intensity can be taken as directly proportional to excited carrier concentration.

Response to comment #1: We thank the referee for giving us a chance to explain this point. Since each excited carrier has a given cross section for generating SEs during its lifetime, the total rate of SE emission is expected to increase linearly with carrier concentration. Experimentally, we find that the intensity of the SUEM contrast images increases approximately linearly as the laser fluence is increased. Since the carrier density is proportional to the laser fluence in the linear absorption regime, this observation heuristically confirms that the SE intensity is roughly proportional to the carrier concentration. We have added a sentence in the revised manuscript to address this point.

Changes made in response to comment #1: On page 2, left column, first paragraph of the revised manuscript, we added a sentence to comment on the relation between the SE intensity and carrier concentration:

“Since each excited carrier has a given cross section for generating SEs during its lifetime, the total rate of SE emission is expected to increase linearly with carrier concentration. This result is consistent with the increased intensity of the SUEM images observed for increasing fluence values.”

Comment #2:

Although the authors also include two movies of the evolution of the SUEM images, I consider Fig. 2 would be improved if similar frame times were included for both n- and p-type samples. i.e. p-type should include 67, 107 and 153 ps frames, and n-type 33, 50 and 127 ps. This would better convey the differences between the evolutions in the article itself.

Changes made in response to comment #2: We thank the referee for this suggestion. We prepared a figure with the electron and hole dynamics at the same delay times, and included it as Figure S1 in the revised Supplementary Information. We considered using this new figure in the main text; however, since we take the signal peak at 50 ps for electrons and 107 ps for holes as the initial time for the simulations, a figure with the same delay times could cause confusion. Hence we decided to keep the initial Fig. 2 in the main text.

We added a sentence on page 3, right column, at the end of the second paragraph, which references the new Figure S1 with the dynamics shown at the same delay times:

“For completeness, contrast images showing the electron and hole distributions at the same set of delay times are given in Figure S1 of the Supplementary Information.”

Comment #3:

The calculation of the second moment via Eqn. (1) should be clarified. This assumes the center of the SE distribution was at $(x,y)=(0,0)$, in conflict with the impression given by Fig. 1. Were $\langle X \rangle$, $\langle Y \rangle$ determined from the data?

Response to comment #3: We thank the referee for making this point. The center of the SE distribution was determined from the data, and we took the center of the distribution to be the origin in Eqn. (1). Note that the center of the distribution does not shift during the diffusive dynamics. We agree with the referee that our choice of the origin of the distribution should be explained, and have added a sentence on this point in the revised manuscript.

Changes made in response to comment #3:

On page 3, right column, below eq. 1, we added a sentence to explain this point:

“The center of the distribution, which is determined from the data and does not shift during the dynamics, is taken to be the origin of the x and y axes in eq. 1.”

Comment #4:

p5 para 2. $\langle R \rangle^2$ should be $\langle R^2 \rangle$.

Changes made in response to comment #4: We thank the referee for this observation. That particular sentence does indeed refer to the variance $\langle R^2 \rangle$. We have corrected this error in the revised manuscript (see page 5, paragraph 2, below eq. 2).

Comment #5:

Eqn. (2) is not an expression for the initial distribution, as stated, but actually describes a time-dependent concentration with Gaussian profile and time-dependent variance. The presentation of this form precedes the discussion of the model which goes on to predict this temporal evolution. Eqn (2) should be an expression for $c(r,0)$ with σ^2 defined as $\langle R^2 \rangle(0)$.

Response to comment #5: We thank the referee for bringing this to our attention. The discussion preceding Eqn. (2) should refer to the equation as the “distribution at all times”, rather than the “initial distribution.” We have corrected this error in the revised manuscript.

Changes made in response to comment #5: On page 5, we changed the sentence before Eqn. (2) to: “The distribution at all times, normalized to the total excited carrier concentration c_0 , reads: ...”

Comment #6:

I am a little unclear regarding the role of the numerical modelling, stated as being used to solve Eqn. (5). This equation has an analytic solution, as mentioned in the text: $c(r,t)$ is given by Eqn (2), with $\sigma^2(t)$ also obtainable analytically given the assumed form for $D^(t)$. Or am I missing something?*

Response to comment #6: We agree that the equation presented in the manuscript can be solved analytically. The numerical solution has been used to verify the analytical solution. The simulations we originally developed included a non-uniform temperature as well as additional effects, and could not be solved analytically. One such simulation with a non-uniform temperature has been included in Figure S2 of the revised Supplementary Information. It considers a non-uniform temperature distribution with a Gaussian spatial profile, which decays exponentially in time and leads to a diffusion equation that cannot be solved analytically. We found that the solution for this more complex model was nearly the same as the solution for the simple model with a uniform temperature distribution presented in the paper. We have added a discussion in the Methods section of the revised manuscript, and present this result in Figure S2.

In response to a comment by another referee, we have additionally carried out ab initio calculations of the bandstructure and density of states in silicon, and employed this data to obtain the average thermal velocity of the electrons and their relaxation time, as well as the initial temperature of the excited carriers in the super-diffusive regime, which is in very good agreement with our previously fitted electronic temperature. These new calculations are presented on page 6 of the revised manuscript. We believe the addition of these results make the modeling part of the manuscript stronger.

Changes made in response to comment #6:

1-- In the Methods section on page 7, right column, first paragraph, we added a sentence: “Note that our model for carrier diffusion can also be solved analytically; the numerical simulations are employed to both validate the analytical solution and to investigate the effect of a non-uniform temperature, which cannot be studied analytically, as described next...” A half-page long discussion on this point follows on page 7, right column.

2-- We added a numerical simulation for a spatially varying temperature with a Gaussian profile. Figure S2 in the revised Supplementary Information compares the second moment obtained for the uniform and non-uniform Gaussian temperature distributions, which lead to second moments and fitting parameters in very close agreement with each other.

3-- A paragraph with ab initio calculations results has been added on page 6, and details of the ab initio calculations are given in the Methods section.

Comment #7:

Why is it appropriate to use the same mobility in the room temperature diffusivity, and that of the excited carriers?

Response to comment #7: As correctly pointed out by the referee, our derivation of eq. 5 neglects the temperature dependence of the mobility. The mobility μ typically decreases with temperature T as a power law above room temperature (e.g., in intrinsic Si, $\mu \sim T^{-2.4}$). We employed samples with a dopant concentration of $\sim 10^{19} \text{ cm}^{-3}$; in such highly doped samples, both electron-phonon (e-ph) and electron-impurity scattering are strong at and above room temperature, resulting in a weak temperature variation of the mobility.

Fig. 5.8. Temperature dependence of mobilities in n-type Si for a series of samples with different electron concentrations. The inset sketches the temperature dependence due to lattice and impurity scattering [5.17]

The figure next to this text (taken from Yu and Cardona, Fundamentals of Semiconductors, Springer, pag. 222) shows the weak temperature dependence of the mobility up to 600 K for Si with a dopant concentration of 10^{19} cm^{-3} . In this figure, and in the conventional theory of charge transport in semiconductors, the temperature is taken to be the *equilibrium temperature of the lattice and carriers*. However, in our work the carrier temperature is very high following laser excitation, but the lattice temperature is significantly lower. The lattice is not excited directly by the

laser; as the carriers cool and give out energy to the lattice through electron-phonon coupling, the lattice temperature increases from its initial room temperature value, but it reaches much lower temperatures than the electrons due to the much higher heat capacity of the lattice compared to that of the carriers. Since the samples do not melt on the surface during the experiment, the maximum lattice temperature never exceeds $\sim 1,000 \text{ K}$ throughout the experiment. In this regime, electron-phonon scattering, and thus the lattice temperature, regulates the mobility. The figure above shows that the mobility varies by less than a factor of 2 up to 600 K, so we believe it is reasonable to neglect the change in mobility compared to the much greater change in electronic temperature T^* in the derivation on page 5 leading to eq. 5. We agree with the referee that this point should be explained more clearly, and have added a sentence in the revised manuscript to justify our assumption.

Response made in response to comment #7: On page 5, right column, below eq. 5, we added a sentence: “Note that we neglect the temperature dependence of the mobility, which is weak for our highly doped samples [26] and controlled by the lattice (as opposed to the much higher electronic) temperature.” Here, ref. 26 is the book by Yu and Cardona mentioned above.

Reviewers' comments:

Reviewer #1 (Remarks to the Author):

I apologized that my poor comments puzzled the authors. The authors emphasize the super-diffusion of the "hot" carriers in semiconductors in the carrier density region of $10^{19-20} \text{ cm}^{-3}$. My naive question is very simple, why the authors focus on the carrier temperature, not scattering time of photoexcited carriers. According to my rough calculation using the free electron mass of m_0 , Fermi energy of $N=10^{20} \text{ cm}^{-3}$ carrier is $E_F = (\hbar^2/2m_0)(3\pi^2 N)^{2/3} = 80 \text{ meV}$, corresponding to $T_F \sim 800 \text{ K}$. The hot carriers in Si is relaxed to $\sim 300 \text{ K}$ within a few ps, experimentally evaluated with photoemission spectroscopy in PRL 102, 087403 (2009). So I think that the carriers with 10^{20} cm^{-3} after the initial relaxation should lie in "dense degenerate region". In this case (quantum regime), carrier-carrier scattering time rapidly increases with the density (for example, PRB 35, 7986 (1987), PRB 43, 7136 (1991)).

The authors considered that the electron-impurity scattering is dominant rather than the electron-hole scattering. The authors reported the slower relaxation of the hot carriers (several tens ps) in this paper. It may imply the suppression of the electron-phonon interaction in heavily-doped semiconductors via Coulomb screening. If so, scattering by the ionized impurities may also be suppressed. It is because the photodoped carrier density (10^{20} cm^{-3}) is much larger than the ionized impurity density (10^{19} cm^{-3}), which is different from the assumption of the Brooks-Herring formula (impurity density = carrier density).

The authors evaluated the diffusion after the initial cooling of the hot carriers. I think that we cannot distinguish carrier cooling and carrier dilution.

The enhancement of the diffusion is very exciting for me, and I agree that this work is valuable for the readers of the Nature Communication. However, carrier density estimated by the authors is high but uncertain ($10^{19}-10^{20}$), so I cannot judge whether this possibility (τ depending on the carrier density) is denied or not. If the authors emphasize the hot carrier effect in their paper, the authors have to deny this possibility before the publication.

Reviewer #2 (Remarks to the Author):

The authors have answered satisfactory all my request and improve the presentation of their results. There is a lot of interest in the understanding and control of dissipation mechanism in low dimensions structures and this work provides a theoretical sound framework to do so. The work is very timing and novel and I think should be published as it is in nature communications.

Reviewer #1 (Remarks to the Author):

Introductory comment: *I apologized that my poor comments puzzled the authors. The authors emphasize the super-diffusion of the “hot” carriers in semiconductors in the carrier density region of 10^{19-20} cm⁻³. My naïve question is very simple, why the authors focus on the carrier temperature, not scattering time of photoexcited carriers.*

Response to comment: In the previous response letter, we addressed the point made by the reviewer, which focused on the effect of carrier density on diffusivity, rather than on scattering times as the reviewer mentions in this new report. We are happy to answer these new comments. We focus on carrier temperature because this quantity can be obtained from the measured diffusivity, as explained on page 5, whereas there is no easy way to measure the scattering time of the super-diffusing carriers in our experiment. Further justification for why theory also focuses on carrier temperature is given in response to comment #1 below.

Comment #1: *According to my rough calculation using the free electron mass of m_0 , Fermi energy of $N=10^{20}$ cm⁻³ carrier is $E_F = (\hbar^2/2 m_0)(3 \pi^2 N)^{2/3} = 80$ meV, corresponding to $T_F \sim 800$ K.*

Response to comment #1: The calculation presented by the reviewer pertains to free electrons *at thermal equilibrium* occupying the bottom of a band with parabolic dispersion and unit effective mass; the occupation assumed in the calculation is 1 up to the Fermi energy, and zero above, so a zero temperature Fermi-Dirac distribution is assumed. The Fermi temperature obtained by the reviewer is thus related to the kinetic energy of a degenerate Fermi gas at zero Kelvin. This situation is not the one relevant in our experiment, where the laser drives the carriers out of equilibrium. Lasers can create carriers with a range of initial non-equilibrium distributions, and these typically relax to a hot Fermi-Dirac distribution on sub-ps timescales due to electron-electron scattering. Given the high temperature of the hot Fermi-Dirac carrier distribution in our experiment, the hot carrier gas is non-degenerate (the average occupation number is much less than 1, different from the scenario described by the reviewer), so that the electrons behave like a classical gas. Our treatment is thus based on a classical gas, with equation of state $PV = nRT$. The relevant variable is the temperature of the excited carriers (more precisely, their energy and momentum distribution) as opposed to the density, because a given density of carriers can be driven to a variety of non-equilibrium distributions, each of which determines different non-equilibrium dynamics.

Comment #2: *The hot carriers in Si is relaxed to ~ 300 K within a few ps, experimentally evaluated with photoemission spectroscopy in PRL 102, 087403 (2009). So I think that the carriers with 10^{20} cm⁻³ after the initial relaxation should lie in “dense degenerate region”. In this case (quantum regime), carrier-carrier scattering time rapidly increases with the density (for example, PRB 35, 7986 (1987), PRB 43, 7136 (1991)).*

Response to comment #2: The paper mentioned by the reviewer (T. Ichibayashi et al., PRL 102, 087403, 2009) employs a pump pulse that results in a much lower

carrier density than in our experiment ($<10^{18}$ cm⁻³, as stated on page 3, right column of the PRL by Ichibayashi et al., vs. $\sim 10^{20}$ cm⁻³ carriers in our experiment at the highest fluence of 1.28 mJ/cm²). We agree that the relaxation time depends on the density, and this is part of the reason why, for example, we find a relaxation time of 77 ps for electrons in our conditions, whereas T. Ichibayashi et al. find a relaxation time of order a few ps. We discuss the origin of our relaxation time on page 6, left column, and in response to comment #3 below.

In the low-density limit, first-principles calculations (Bernardi et al., PRL 112, 257402, 2014) show indeed that hot carriers generated by ~ 2 eV light thermalize in roughly a few ps in Si, consistent with the paper mentioned by the reviewer. While we agree that the relaxation time depends on density, this density dependence is irrelevant in our experiment, since, for each given fluence the average carrier density is nearly constant throughout our measurement. With a much lower density one would observe super-diffusion for a shorter time, thus making the process more challenging to measure.

To address the reviewer's comment, we have added a note in the revised manuscript explaining that the regime studied here is distinct from the low-density regime, in which hot carriers thermalize in a few ps. The papers by Ichibayashi et al. and Bernardi et al. have both been cited.

Changes made in response to comment #2: On page 6, left column, we added the following sentence: "Note that the high carrier density regime probed here is distinct from the low density regime, in which hot carriers thermalize in a few ps [10,27]." Here, refs. 10 and 27 are the papers by Bernardi et al. and Ichibayashi et al., respectively.

Comment #3: *The authors considered that the electron-impurity scattering is dominant rather than the electron-hole scattering. The authors reported the slower relaxation of the hot carriers (several tens ps) in this paper. It may imply the suppression of the electron-phonon interaction in heavily-doped semiconductors via Coulomb screening. If so, scattering by the ionized impurities may also be suppressed. It is because the photodoped carrier density (10^{20} cm⁻³) is much larger than the ionized impurity density (10^{19} cm⁻³), which is different from the assumption of the Brooks-Herring formula (impurity density = carrier density).*

Response to comment #3: In our discussion on page 6, we explain that the relaxation time found here is the time for the sample to return to thermal equilibrium after excitation with the laser pulse; after such equilibration, the sample exhibits the typical room temperature steady-state diffusivity. The fact that SUEM probes the high-energy tail of the distribution, and the fact that phonons also need to return to equilibrium for the sample to exhibit the room temperature diffusivity (essentially, at these high densities one has hot phonon effects), both contribute to determine the diffusivity decay time constant (i.e., the hot temperature "relaxation time") found in our work. Also note that the decay time constant of the diffusivity, which is the timescale reported here, is related, but not exactly equivalent, to the electron relaxation time measured in a pump-probe experiment. We agree with the reviewer that screening of the electron-phonon coupling may also be responsible for the longer relaxation time, and note this in the revised manuscript.

Changes made in response to comment #3: On page 6, left column, we added a sentence to comment on the origin of the longer relaxation time: “Different from the low density regime, both hot phonon effects and screening of the electron-phonon interaction can contribute in our experiment to increasing the diffusivity decay time constant.”

Comment #4: *The authors evaluated the diffusion after the initial cooling of the hot carriers. I think that we cannot distinguish carrier cooling and carrier dilution.*

Response to comment #4: The “dilution” of the carriers, as the reviewer puts it, stems from the spatial dependence of the carrier concentration, which gives rise to the diffusion process visualized in our experiment through the SUEM images. The energy distribution of the carriers, and thus their effective temperature, is a rather different matter, since one could in principle achieve any combination of carrier temperature and spatial concentration profile. We agree that the charge carriers cool while they diffuse in our work, a result that is clearly explained in the manuscript and that gives rise to a time-dependent transient diffusivity. This is indeed the central result of our work (for example, see equations 4 and 5 on page 5).

Comment #5: *The enhancement of the diffusion is very exciting for me, and I agree that this work is valuable for the readers of the Nature Communication.*

Response to comment #5: We thank the reviewer for stating that our results are very exciting and valuable for the readers of Nature Communications, and are grateful for the additional feedback. We hope that the responses provided herein and the changes made to the manuscript help address any remaining concern.

Comment #6: *However, carrier density estimated by the authors is high but uncertain (10^{19} - 10^{20}), so I cannot judge whether this possibility (τ depending on the carrier density) is denied or not. If the authors emphasize the hot carrier effect in their paper, the authors have to deny this possibility before the publication.*

Response to comment #6: We believe that the relaxation time dependence on the density, as noted above, is not of main relevance for our results: the average carrier density is nearly constant throughout our experiments, yet the diffusivity varies by 4 orders of magnitude. Thus, any density dependence of the relaxation time, even if present, would not affect the observed super-diffusion at any given fluence value. The *average* carrier density at our fluence is, to be precise, $8 \times 10^{19} \text{ cm}^{-3}$. This value is obtained using the known value of the fluence (1.28 mJ/cm^2), the photon energy of 2.4 eV, and the absorption coefficient of Si at 515 nm, which is given in Ref. 21. The initial carrier density is given as a range in the manuscript since the beam has a Gaussian profile, so the density is not uniform. We have changed the manuscript to provide the precise average value of the initial carrier density as opposed to the 10^{19} – 10^{20} cm^{-3} range as we did in the previous version.

Changes made in response to comment #6:

1) We added a sentence on page 6, left column: “While the relaxation time generally depends on carrier density, this dependence does not affect our measured transient

diffusivity, since for each given fluence value the average carrier density is nearly constant throughout our experiment.”

2) The initial average carrier density at our fluence, $8 \times 10^{19} \text{ cm}^{-3}$, is given in the revised manuscript on page 4, right column.

Reviewer #2 (Remarks to the Author):

Comment: *The authors have answered satisfactory all my request ad improve the presentation of their results. There is a lot of interest in in the understanding and control of dissipation mechanism in low dimensions structures and this works provides a theoretical sound framework to do so. The work is very timing and novel and I think should be published as it is in nature communications.*

Response to comment: We are grateful for the very positive feedback.

REVIEWERS' COMMENTS:

Reviewer #1 (Remarks to the Author):

I considered that their experimental results were interesting, but that some interpretation was still arguable in the previous reviewing. However, the authors have answered my questions and modified some description. I consider that this manuscript is suitable for the publication in Nature Communications.

Reviewer #2 (Remarks to the Author):

The authors have answered very convincingly all referee's concerns. I found the work suitable for publication in nature communications.

POINT-BY-POINT RESPONSE TO THE REVIEWERS' COMMENTS

REVIEWERS' COMMENTS:

Reviewer #1 (Remarks to the Author):

I considered that their experimental results were interesting, but that some interpretation was still arguable in the previous reviewing. However, the authors have answered my questions and modified some description. I consider that this manuscript is suitable for the publication in Nature Communications.

Reply to comment: We are delighted that the reviewer is satisfied with our revisions. Thank you for recommending our paper for publication in Nature Communication.

Reviewer #2 (Remarks to the Author):

The authors have answered very convincingly all referee's concerns. I found the work suitable for publication in nature communications.

Reply to comment: We are glad that the reviewer found our answers convincing. Thank you for stating that our work is suitable for publication in Nature Communication.